# Scaling for African Inclusion in High-Throughput Whole Cancer Genome Bioinformatic Workflows

**DOI:** 10.3390/cancers17152481

**Published:** 2025-07-26

**Authors:** Jue Jiang, Georgina Samaha, Cali E. Willet, Tracy Chew, Vanessa M. Hayes, Weerachai Jaratlerdsiri

**Affiliations:** 1Ancestry and Health Genomics Laboratory, Charles Perkins Centre, School of Medical Sciences, Faculty of Medicine and Health, The University of Sydney, Camperdown, NSW 2050, Australia; jue.jiang@sydney.edu.au; 2Sydney Informatics Hub, The University of Sydney, Camperdown, NSW 2050, Australia; georgina.samaha@sydney.edu.au (G.S.); cali.willet@sydney.edu.au (C.E.W.); tracy.chew@sydney.edu.au (T.C.); 3Manchester Cancer Research Centre, The University of Manchester, Manchester M20 4GJ, UK; 4School of Health Systems and Public Health, University of Pretoria, Pretoria 0002, South Africa; 5Norwich Medical School, University of East Anglia, Norwich NR4 7TJ, UK; 6Computational Genomics Group, Charles Perkins Centre, School of Medical Sciences, Faculty of Medicine and Health, The University of Sydney, Camperdown, NSW 2050, Australia

**Keywords:** Africa, computational workflow, parallelism, cancer genomics, whole-genome sequencing

## Abstract

Africa faces the highest mortality rates across eight cancer types. However, cancer studies are biased toward European populations, leading to major concerns that cancer treatments may be ineffective for African patients. Providing a systematic review of African-inclusive whole cancer genome studies, African-derived tumours reveal distinct clinically relevant drivers, molecular taxonomies, and overall increased genomic instability, highlighting challenges associated with non-African-derived computational workflows. We provide a rationale for parallelism strategies to accelerate the processing steps of those distinctly intensive data, allowing for required scalability. Advocating for further resources that capture the rich African ancestral diversity, a concerted global effort will be required to improve and ultimately standardise bioinformatic workflows, thereby enhancing health outcomes for African cancer patients.

## 1. Introduction

Sub-Saharan Africa bears a disproportionate burden of many cancer types, as reported in GLOBOCAN 2022 [1]. In addition to cancers with a known viral aetiology, such as cervical uteri cancer and Kaposi sarcoma, Africa has the highest age-standardised mortality rates (per 100,000 people per year) for eight other cancer types: breast (19.2), prostate (17.3), non-Hodgkin lymphoma (3.3), thyroid (0.64), vulva (0.63), Hodgkin lymphoma (0.41), salivary glands (0.38), and vagina (0.24) [1]. Despite this burden, access to tailored clinical care remains restricted. This is largely attributed to limited tumour genome profiling resulting in lack of population-relevant data required to enable precision medicine implementation [2,3,4], compounded by inadequate investment, resources, and technical capacity [5].

Cancer whole-genome research conducted across high-income countries has inevitably exhibited ancestral bias [5]. While patients of European ancestry predominate, African ancestral representation is largely limited to African American individuals. Consequently, ancestral fractions are biased towards West African origins, further obscured by admixture with European ancestry (on average, 18%) [6], limiting cross-continental correlations. This focus on African American populations applies not only to studies of a particular cancer type, taking prostate cancer (PCa) as an example [7,8,9,10,11,12,13,14,15], but also to pan-cancer studies. The largest of these, Pan-Cancer Analysis of Whole Genomes (PCAWG), is derived from merging efforts generated by the UK-led International Cancer Genome Consortium (ICGC) [16] and the US-led Cancer Genome Atlas (TCGA) [17]. Among the 2583 tumour-normal matched whole-genome sequences (WGSs) across 38 cancer types, only 5% were of African ancestry (African ancestral fraction: median, 83.91%; range, 50.02–99.96%) [18]. Another study of 333,908 tumour-only samples across six cancer types included 9.8% of patients of African ancestry (unknown African ancestral fractions) [19]. As such, regions of Africa most impacted by cancer mortality, such as southern and central Africa for PCa and northern, eastern, and southern Africa for non-Hodgkin lymphoma [1], are unlikely to fully benefit from targeted cancer care built on these datasets. Nevertheless, tumours derived from African American patients show molecular differences compared to European patients. For example, pan-cancer databases show that tumours derived from African American patients have significantly elevated rates of whole-genome duplication (WGD) [20], as observed in PCAWG (n = 1293, *p*-value = 0.034) and TCGA (n = 8060, *p*-value = 0.022), and further validated by the Memorial Sloan Kettering—Metastatic Events and Tropisms (MSK-MET) array-sequenced study (n = 13,071, *p*-value = 0.016). The increased WGD may be partly attributed to the higher frequencies of *TP53* mutations and cyclin E (*CCNE1*) gain in African American patients (*p*-value = 5.8 × 10^−7^ and 2.5 × 10^−5^, respectively) [20]. Additionally, TCGA through whole-exome sequencing reported a higher level of intratumor heterogeneity (ITH) in African American breast cancer (BRCA) patients (n = 768) by 5.1 units calculated using the mutant-allele ITH algorithm [21]. In contrast, lower frequencies of *TMPRSS2–ERG* gene fusions and *PTEN* losses have been noted in African American PCa patients (n = 24; 21% versus 40–80%, 8% versus 40%, respectively) [13]. These differences highlight the importance of including regionally diverse populations across the African continent. Leveraging large-scale whole-genome tumour data, cancer discoveries can be extended to critical non-coding and more complex structural cancer drivers, mutational signatures, and molecular subtyping to reveal potential aetiologies specific to African patients.

Large-scale studies involving whole tumour genome interrogation face computational challenges in processing workflows. PCAWG reported a total of 10 million CPU-core hours used for their workflows [18]. Generating and analysing WGS data in the Binary Alignment Map (BAM) format (each ~100 GB for a 30X genome) requires substantial computational resources, including large storage hardware, multiple CPUs, and high memory allocation. Such demands increase proportionally with cohort size, incurring additional computational costs associated with achieving greater statistical power and sensitivity. For example, to accurately identify short variants—single nucleotide variants (SNVs) and insertion/deletion (indel) variants less than 50 bp—PCAWG employed six different algorithms to produce a consensus call set [18]. Notably, the Genome Analysis Toolkit (GATK) pipeline includes a joint-calling step to incorporate cohort-wide information [22,23]. Such resource requirements can only be met by high-performance computing (HPC) platforms [24,25], supporting job parallelism and allocating hundreds of CPUs and terabytes (TB) of random-access memory (RAM) in a single run. The scatter–gather approach proposed by the GATK pipeline [22] enables the execution of several thousand parallel tasks in a cost-effective and fast manner. It is a higher level of parallelism than the conventional parallel-by-sample strategy, allowing for simultaneous execution of multiple tasks divided from a single step of processing a sample. Integrating high-level parallelism for the interrogation of African WGS data using HPC platforms would accelerate the pace of research and as such greatly contribute to closing the gap in African genomic inclusion.

We first examined all publicly available WGS databases that include tumours and matched blood or normally derived tissue from cancer patients from Sub-Saharan Africa. We highlighted ancestry-related molecular features and bioinformatic tools used across studies for genome alignment and variant calling. We showed the scientific importance and computational demands of analysing African cancer genomes. To alleviate the computational burden and enhance the efficiency, we presented a scalable bioinformatics workflow deployed on HPC infrastructure. The scalability, defined as maintained time- and CPU-efficiency even for large-scale cohorts, is achieved by high-level parallelism through physical data or genomic interval chunking strategies applied to computationally intensive steps. Evaluations and improvements of computational performance of these steps were benchmarked and tested using African WGS data. Furthermore, we discussed the potential improvements and applications of introducing new genomic technologies.

## 2. WGS Data of African Patient-Derived Tumours

Through the literature review using PubMed with the following search terms—‘WGS’ or ‘whole-genome’, ‘cancer’ or ‘tumour’ or ‘carcinoma’, ‘Africa’ or ‘African’ or ‘African descent’ or ‘Sub-Saharan’, ‘patients’ or ‘cohort’—a total of 154 publications were identified on 3 June 2025. After selecting studies with patients from any of the 43 countries across Sub-Saharan Africa, we identified seven publications from five consortia that analysed WGS datasets derived from tumour–blood patient-matched samples, as summarised in Table 1 and Figure 1. For each consortium, we briefly reviewed (i) the cohort information, such as the countries and cohort size; (ii) the reported biological findings; and (iii) the workflow used for analysing the WGS data.

### 2.1. Cohort Information of African Patients

The Southern African Prostate Cancer Study (SAPCS) expanded from an initial cohort of six Black South African patients [27] to include a total of 118 genetically defined African ancestral PCa cases [26]. More recently, SAPCS has merged with partner studies as part of the Health Equity Research Outcomes and Improvement Consortia (HEROIC) Prostate Cancer Precision Health (PCaPH) Africa1K initiative [33]. In East Africa, three studies have emerged, two focusing on esophageal squamous cell carcinoma (ESCC) [28,29] and one on Burkitt lymphoma (BL) [30]. The largest study cohort for Sub-Saharan Africa, the Oesophageal Squamous Cell Carcinoma African Prevention Research (ESCCAPE) study, included 162 patients from Kenya, Malawi, and Tanzania [28]. The second ESCC study, conducted under the African Esophageal Cancer Consortium (AfrECC), recruited 61 patients from Tanzania [29]. The Burkitt Lymphoma Genome Sequencing Project (BLGSP), including the Epidemiology of Burkitt’s Lymphoma in East African Children and Minors (EMBLEM) study, focused on the role of the Epstein–Barr virus associated with BL [30,31]. This project initially conducted WGS sequencing on samples from 74 patients and later expanded to 87 patients. In West Africa, the Nigerian Breast Cancer Study (NBCS) generated whole-genome data from 97 Nigerian women diagnosed with BRCA [32].

### 2.2. Cancer Discoveries from African Genomic Studies

We further interrogated genomic features that are predominant in African patients and shared across cancer types, as summarised in Table 2. SAPCS and NBCS included clinicopathologically matched European ancestral patients to provide direct clinical, technical, and informatic comparative analysis, while NBCS further included an African American cohort for comparison. In contrast, ESCCAPE, AfrECC, and BLGSP did not include direct ancestral comparisons. For this reason, we compared the ESCCAPE and BLGSP results by the country, where the ancestries of patients were not determined, and compared the AfrECCE results referred to external publications. BLGSP reported that Epstein–Barr virus (EBV) infection in BL showed a higher mutational burden and more aberrant somatic hypermutation, which could also associate with ancestral or geographical factors (e.g., higher exposure to EBV) given that EBV-positive patients were largely Ugandan (68 out of 71, 96%). Overall, African-derived tumours presented with significantly more variants, ranging from short to structural variant types, with elevated frequencies and longer-tail of cancer driver mutations. In contrast, African-derived prostate tumours showed a diminished frequency for *TMPRSS2–ERG*, which is common for European patients, and ESCC tumours showed a decreased frequency for *TP53* mutations, although it remained the top candidate driver.

As cancer drivers are implicated in promoting tumour initiation and progression, African patients presenting distinct mutational patterns may undergo a unique evolutionary trajectory triggered by previously unrecognised aetiologies. SAPCS identified two African-specific mutational subtypes in PCa: one predominated by driver gene copy number (CN) gain and included enrichment for driver mutations in *KMT2C*, *MTOR*, and *TP53* among inferred tumour subclones, while the second demonstrated a combination of CN gain and hemizygous loss in cancer drivers. Further studies using SAPCS data found that the aggressive presentation of prostate tumours, defined as the International Society of Urological Pathology (ISUP) ≥ 3, was significantly associated with other molecular features for African patients. This includes type-specific hyper-SV subtypes [37], shortened tumour telomere lengths against leucocyte-derived lengths [38], and megabase impacting Y-chromosomal CN gains over losses [39]. NBCS reported a molecular subtype of BRCA featured by an African-related cancer driver, *GATA3*, at the early clonal stage, with a 10.5-year early diagnosis, and a novel aetiology-unknown signature (INDEL-B) strongly associated with African ancestry. Investigated by ESCCAPE, smoking and alcohol consumption are known factors for ESCC, but their associated genomic signatures were not identified in patients from Africa. Likewise, AfrECC showed no association with smoking and African relevant RNA-derived subtypes. Together, these findings reveal a spectrum of African relevant mutational patterns largely lacking known aetiologies or established clinical implications. This highlights an urgent need for African-inclusive studies to investigate underlying risk factors with comprehensive clinical follow-up data and a sufficient cohort size to achieve statistical power.

### 2.3. Challenges of Analysing WGS Data of African Patients

African cancer study workflows described above have mostly followed the same pipeline architecture from read alignment to variant detection, with utilised tools listed in Table 3. For read alignment (or read mapping) to a known/reference human genome, all studies used the BWA-MEM aligner [40]. The aligned reads, stored in a BAM file format, are used for subsequent variant detection, with studies employing different tools. The choice of variant calling tools is known to impact the sensitivity, accuracy, and reproducibility of the results [41], as well as computational resource requirements and scalability for large cohort consideration.

The computational challenges of analysing African-derived WGS data from large-cohort studies stem from three aspects: the large size of the WGS data, the methods adopted for variant calling with enhanced sensitivity, and the elevated mutational burden of African-derived tumours. Firstly, WGS data of tumour and patient-matched normal samples typically require a minimum of 60X and 30X sequencing coverages, respectively, with an average size of 300 GB per patient in SAPCS. High coverage, demanding extensive time for alignment and analysis, benefits downstream analyses, such as clonality interrogation which is essential for studying cancer development. The SAPCS, for example, spent a total of 712,200 service/compute units in HPC servers to process 190 patients, of which 118 were African. Secondly, the computational burden is exacerbated by leveraging cohort-wide information and multiple-caller adoption for the sensitivity of variant calling. For germline short variants, the HaplotypeCaller employed by SAPCS used 16,500 service units for joint calling, which exclusively allows for genotyping at the cohort level without any sample size restriction (a maximum of ten for Strelka2). Joint calling reduces false negatives by enhancing the detection of common variants within samples that may be affected by quality issues at each genomic position; reduces sequencing errors falsely called as variants by downgrading the confidence of calls in one sample that are invariant in all others; and provides genotype consistency which is difficult to attain when merging single-sample variant data. For somatic variants, SAPCS (40,700 service units used) and NBCS created a panel of normal (PoN) to filter out false positives caused by germline variants and artefacts raised from sequencing and data processing. Similarly to the PoN strategy, BLGSP filtered out a set of SVs that were called in multiple samples. In addition, consensus call sets merged from several callers have been adopted for somatic short variants by ESCCAPE, BLGSP, and NBCS, as well as for SVs by SAPCS, BLGSP, and NBCS. Lastly, compute time is longer for African patients with higher genomic instability. Using SAPCS data, we found longer execution hours for African data than European data when performing GRIDSS for SV detection (median, 11 versus 9.6 h; *p*-value = 0.0002). These computational burdens are expected to be exacerbated with expanding cohort size, highlighting a need for scalable and well-optimised workflows.

## 3. Rapid and Scalable HPC Workflow for African Genomic Studies

Aiming to meet substantial computational demands while improving rapid WGS processing time and aligning resource usage with underlying computer hardware, SAPCS has reported adaptive pipelines for rapid and scalable processing on HPC platforms. Here, we provide a closer evaluation of the SAPCS workflow by briefly introducing (i) steps of processing WGS data, (ii) the parallelism strategies applied to computational-intensive steps, and (iii) describing more recent improvements.

### 3.1. SAPCS Workflow Overview

SAPCS applied a parallelism-integrated workflow (code/scripts available online) [55,56,57] on African WGS data adapted to HPC infrastructure. Ideally, the workflow could finish processing any size of the cohort in two days, if ignoring the queue time of the HPC server and allowing enough computational resources. The modular workflow applies physical data chunking to the most compute-intensive phase of read mapping and genomic interval chunking to the GATK Best Practices workflows for germline and somatic variant detection [22,23]. The workflows consist of four pipelines from data pre-processing to variant identification, as presented it Figure 2. Analysis-ready BAM files are prepared in Pipeline 1 for variant discoveries, including short germline variants, short somatic variants, and SVs, which are processed in Pipelines 2 to 4, respectively. Using real-world SAPCS data, we benchmarked the optimised resource configurations on Australia’s National Computational Infrastructure (NCI) Gadi HPC, with performance summarised in Appendix A and determined the best batch-processing configuration for high-level parallelism steps with a total execution time within two days, presented in Table 4.

Pipeline 1 processes raw WGS data from each blood and tumour sample in the FASTQ format into analysis-ready data in the BAM format, which contains information about the aligned coordinates of reads on the human reference genome. The ALT-aware function of BWA-MEM extends the mapping region from the primary human reference genome GRCh38 to a list of alternative haplotypes derived from broader populations, thereby expanding the investigation of immune regions among African patients. However, reads aligned with multiple regions may receive low mapping quality scores, requiring manual checking during variant calling. Without high-level parallelism, the read alignment required 15.1 h for a 30X coverage WGS data (6-CPU, 24 GB RAM allocation). Therefore, the pipeline performs alignment through physical data chunking, enabling the parallelisation of large, multi-node jobs with reliably high CPU efficiency and predictable execution time. The generated scattered alignments are merged into one BAM file per sample utilising SAMBAMBA. The following refinement, which facilitates the utmost sensitivity and specificity of variant calling, includes masking duplicate reads (technical artefacts that may cause false positives) utilising SAMBLASTER and GATK base quality score recalibration (BQSR) to lessen the impact of systematic errors introduced during sequencing. The GATK package was overhauled with version 4 to replace multi-threading functionality for resource-intensive tasks with scatter–gather capability via the ‘intervals’ flag. The final quality control stage examines mapping quality, sample contamination, and tumour cellularity using Qualimap, Sequenza, and QSignature, respectively.

Pipeline 2 identifies germline short variants using BAM files from blood samples (or normal tissue) generated in Pipeline 1, employing variant calling, joint-calling, and filtration stages. Samples are first processed with GATK HaplotypeCaller by 3200 genomic intervals, followed by the joint-calling stage to enhance variant detection sensitivity, as previously discussed. To reduce the memory demand of joint-calling, we utilised intervals enabling 3200-fold parallelisation per cohort and the GenomicsDB format that deals with the cohort-wise variant data, followed by merging all intervals to one cohort-level variant file via GatherVcfs. The following filtration stage of variant quality score recalibration (VQSR) applies machine learning algorithms to assess the pattern of known validated variants (provided in the form of reference SNP and indel databases) from the cohort-level variant file, which estimates the trustworthiness of all variants. To ensure sufficient data for the model training, VQSR does not employ any chunking strategies.

Pipelines 3 and 4 identify somatic short and SVs, respectively. Somatic variants are those present only in a tumour sample and absent in the matched blood (or normal when applicable) sample, so the identification process takes BAM files from paired tumour and blood samples as inputs. Pipeline 3 first creates a PoN to enhance variant specificity as previously described. Similarly to the strategy applied in Pipeline 2, the PoN data are generated in 3200 genomic intervals, transformed into the GenomicsDB format, and merged into a single cohort-level file. The PoN is included in the variant calling stage performed by Mutect2. The Mutect2, with its improvement in detecting low-frequency variants, facilitates the investigation of cancer subclonal evolution. The last filtration stage with FilterMutectCalls excludes several types of artefacts fitted by models such as those introduced by formalin fixation (although not necessary for the fresh tissue from SAPCS).

Pipeline 4 is more complicated and computationally intensive than short variant detection. This is because SVs can involve thousands to millions of base pairs, span multiple chromosomes, and often have very complex forms, including deleted or inverted sequences, chromosomal translocations, or combinations of different SV types. Due to the inescapable fact that different types of SVs are called with varying accuracy using different algorithms [63], SAPCS adopted GRIDSS and Manta callers to find a consensus call set. These callers require access to the entire dataset of a sample (tumour and matched blood samples), so data or interval chunking is not possible but parallelised by the sample.

### 3.2. High-Level Parallelism

For African ancestry WGS presenting elevated germline and somatic variants, its workflow needs to be scalable for improved execution time efficiency and, therefore, the analysis of a larger cohort size. The pipelines described above have been tailored to accommodate two types of high-level parallelism: physical data chunking for read alignment and the scatter-gather of genomic interval processing for variant calling. The execution time of these steps has been improved substantially. Using hundreds of computational cores, the execution time of the alignment has been reduced from 15.1 h to 2 h for a 30X sample, and the germline haplotyping step improved from 7.8 h to 0.5 h. The somatic variant calling pipeline took approximately 3.3 h for a cohort of 20 patients, compared to 36 to 47 h using the pipeline employed by the Pan Prostate Cancer Group (PPCG) consortium (https://github.com/cancerit/dockstore-cgpwgs accessed on 9 June 2025; 48 CPUs and 960 GB each).

#### 3.2.1. Parallelism via Physical Data Chunking for Alignment

Using SAPCS African data, we show that alignment time is improved to less than two hours through physical data chunking (or sharding) of input reads while maintaining mapping outcome due to the independent alignment of each sequencing read. The paralleled alignment stage is achieved by three steps: (i) splitting FASTQ inputs into small and independent files of homogenous size; (ii) mapping small files in parallel; and (iii) merging BAM alignment files, as shown in Figure 2. Around one hour was expected to split input FASTQ files into about 184 pairs (forward and reverse reads) of small files that each contained two million reads and took five minutes for alignment by BWA-MEM (6 CPUs). The BWA-MEM mapping showed high and consistent performance over 0.87 and CPU efficiency over 0.83 throughout the compute allocations from one to eight nodes (48 CPUs per node) as shown in Figure 3a. For an 80-node allocation job processing 20 blood samples (~30X coverage each; 3840 parallel tasks expected) at once, the batch was completed in 0.53 h (32 min) with high CPU efficiency (0.84). The outputs of mapping—scattered BAM files—took around 0.3 h to merge per blood sample (~30X coverage) and twice the time for tumour samples attributed to a doubling of sequence coverage (~60X coverage), as shown in Table 4.

#### 3.2.2. Parallelism via Genomic Interval Chunking

Genomic interval chunking, also known as scatter–gather by GATK, is a parallelism strategy developed particularly for bioinformatics analysis. The human reference genome should be partitioned into evenly sized, abutting intervals. Each interval is processed independently in parallel. The strategy is applied in Pipelines 1–3 with varying numbers of genomic intervals to optimise execution time and computational load without impacting outcome. For the BQSR stage in Pipeline 1, the recalibration implements machine learning models of known variants to estimate a variant’s quality, so the step was parallelised into 32 interval tasks to allow for adequate training data per interval for the recalibration model. The following step of applying the recalibration is not computationally intensive and is parallelised into 24 intervals. In contrast, the 3.2 billion nucleotide-long human genome was divided into 3200 intervals to computationally intensive steps in Pipelines 2 and 3, such as the variant calling of local re-assembly of DNA haplotypes via HaplotypeCaller for germline variants and MuTect2 for somatic variants.

We assessed the scalability of HaplotypeCaller and Mutect2 using scaling tests and batch processing with African SAPCS data. HaplotypeCaller maintained performance over 0.80 and CPU efficiency over 0.77 when using one or two compute nodes, and the CPU efficiency decreased when allocating more than two nodes (3200 parallel tasks for a 30X blood sample), as depicted in Figure 3b. The CPU efficiency was affected by idle CPUs caused by varying execution time of parallel tasks which depends on the local read depth and the unpredictable number of potential variants per interval. Scaling to process 20 blood samples at a single run, the batch job was completed in 1.8 h and maintained a 0.98 CPU efficiency with a 20-node allocation. Additionally, MuTect2 performs two steps of the Pipeline 3 somatic variant discovery pipeline, including creating a PoN and performing variant calling. The PoN creation step showed good scalability, performance over 0.88 and CPU efficiency over 0.9 when allocated from one to six nodes (3200 parallel tasks for a blood sample), as depicted in Figure 3c. Consistently high CPU efficacies of MuTect2 were also shown for the batch processing of PoN (20 blood samples, 64,000 parallel tasks) and variant calling steps (20 pairs of tumour and matched blood samples, 64,000 parallel tasks), which completed within one hour (0.58 and 0.81 h, respectively).

### 3.3. Integration with Workflow Management Tools

While parallelisation methods dramatically improve the performance of computationally intensive processes within a workflow, the real-world implementation of large-scale WGS workflows also requires robust orchestration to manage thousands of tasks and ensure reproducibility. Manual submission of batch jobs in HPC environments with strict wall time limits and diverse job profiles introduces inefficiency. While steps employing high-level parallelism are optimised to have similar execution times for each parallel task by ensuring even input sizes or genomic intervals, other steps that batch-process multiple samples and utilise parallelism only at the sample level can suffer from idle CPU time due to the varying execution times between parallel tasks. Reducing manual manipulation and idle CPUs could be achieved simultaneously by introducing workflow management tools, such as Nextflow [64]. Instead of batch processing, Nextflow enables the independent processing of multiple samples which could decrease the idle CPU time by automatically assigning idle CPUs to tasks, as exemplified by FASTQ splitting and BAM merging steps in the alignment, as illustrated in Figure 4.

## 4. Emerging Technologies and Resources to Be Integrated to African Genomic Studies

Although the diversification of African biospecimen collections is gaining attention, it is still drastically insufficient, while analytic methods and reference resources remain African-exclusive in genomic applications. The application of new genomic technologies brings promises of improvements in both cancer research and clinical implications, as summarised in Figure 5. The emerging long-range sequencing/non-sequencing technologies can facilitate SV detection. The optical genome mapping (OGM) or digital karyotyping method is designed to capture single molecules up to megabases [65], as presented in Figure 5a. OGM provides a cost-effective option to detect and visualise SVs requiring no bioinformatic skills due to the user-friendly Bionano platform. As such, OGM is suitable for clinical use, such as a quick preliminary screening for progressive tumours to determine the necessity of in-depth investigation of SVs. For cancer research, OGM and long-read sequencing have validated tumour DNA sequences impacted by SVs [26,37]. Complementary to OGM, long-read sequencing enables base resolution with phasing information and allows for an extended search in low-complexity regions [66]. A recent study suggests a link between the early diagnosis of cervical cancer in African American women with *YAP1* amplification and the *YAP1–BIRC3–BIRC2* breakage–fusion–bridge cycle identified from cell lines using long-read sequencing [67]. Due to the high cost of long-read sequencing and importantly the need for intact high molecular weight DNA, this technology has not yet been applied for large-scale cancer studies even among European patients.

Being aware of genetically higher diversity in Sub-Saharan African populations, we urge the need for African-adapted resources to remove bias in genomic research. Long-range sequencing technologies can contribute to a more comprehensive genome assembly. The growing recognition of diversity and inclusion in human genetics has led to widespread calls for improving methods for presenting global variation. Recently, using the DNA technologies described above to include gapless telomere-to-telomere (T2T) assemblies, a truly complete genome for an individual has been constructed, as illustrated in Figure 5b. For example, a complete hydatidiform mole (CHM13) has filled 8% gaps in GRCh38, although biased towards European ancestry [69,70,71]. Following this, although not gapless, the Human Pangenome Reference Consortium (HPRC) released a pangenome draft built from 47 subjects, including four from Sierra Leone, three Nigerians, a Kenyan, and a Gambian [72]. The pangenome draft has reported additional African-specific SVs that are related to epigenetic features [72]. New bioinformatic methods are urgently needed to refine each T2T assembly from genetically diverse individuals for the real-world use of this advanced pangenome reference concept. Relevant tools designed for the application of emerging T2T and pangenome references are still developing [73] and would shift the current paradigm in cancer genomics analyses if successfully implemented. These population-aware efforts in previously under-ascertained regions of the human genome would pave the way for generating practical and translational insights from cancer genomics studies in Africa, as illustrated in Figure 5c.

While OGM cannot exclude for sequencing, as it is unable to detect small variants, both OGM and long-read sequencing technologies are limited by their dependence on acquiring high or ultra-high molecular weight DNA. Unsuitable for use on highly abundant, yet highly degraded, formalin fixed tissue, these technologies require a higher quality of tissue sources and abundance, as well as efficient laboratory skills to acquire intact kilo-to-megabase-long DNA molecules. The latter highlights the need for effective biobanking of fresh tissues across Africa to facilitate future application of these emerging technologies.

In addition, Sub-Saharan African representatives are limited in currently available large-scale resources, such as the Human Genome Diversity Project (HGDP) and the 1000 Genome Project (1KGP) [74]. These resources are involved in variant filtering steps and could affect downstream analyses, such as cancer drivers and signatures. A more suitable resource for African research is to use a variant set generated from a panel of young and healthy African individuals, to counteract any geographical and ancestral differences. This resource can serve as a PoN for variant filtering, especially for tumour-only variant calling, although the potential benefits remain undetermined.

## 5. Conclusions and Challenges

Sub-Saharan African countries have demonstrated greater risks of many cancer types than countries of other continents. However, cancer genomic studies, especially large-scale studies, often lack representative data from Sub-Saharan Africa. We report published research on tumour WGS data derived from African cancer patients, revealing only five databases representing four cancer types. Reviewing genetic findings from these studies, unique molecular patterns within tumours derived from African patients have been observed when compared to European patients. These include higher genomic instability, varying frequencies for cancer drivers, and a diversity of tumour subtypes with unknown underlying mechanisms. Although limited by the small number of studies, the findings support a pressing need to strive for African-inclusive cancer research to facilitate equitable patient care and outcomes. Exploring the WGS bioinformatic workflow implemented in these limited African-focused cancer studies, we describe computational barriers. Due to the challenges, we introduce a highly scalable, efficient and rapid workflow that outlines how modern computing techniques, combined with appropriate access to computing hardware, can meet the computational burden for large-scale African inclusive cancer studies. We acknowledge that future studies are required to determine variant calling accuracy of the tested pipeline. The analysis should compare the performance on a “truth set” of African ancestry and test the improvements of integrating cohort information. Beyond short-read WGS data, emerging genomic technologies offer more accurate options that could be applied in future research and clinical use, from generating African-representative T2T-finshed pangenome references to addressing the heightened genomic complexity observed across these limited African cancer genome studies. While technologies may reduce the need for highly skilled computational biologists, they will require high-quality intact DNA, highlighting the need for concerted efforts to expand fresh tissue biobanking across Africa. With improved computing practices that scale efficiently to large cohorts and advanced technologies that provide unprecedented genomic resolution, these combined efforts could help progress genomic applications in cancer diagnosis and treatment in Sub-Saharan Africa.

## Figures and Tables

**Figure 1 cancers-17-02481-f001:**
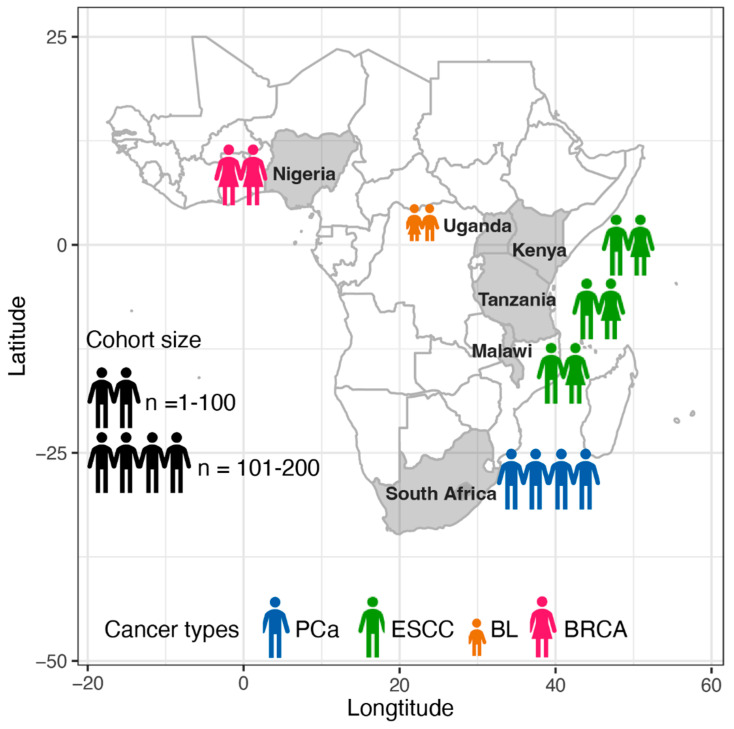
Cancer genomic databases established in Sub-Saharan Africa. PCa, prostate cancer; ESCC, esophageal squamous cell carcinoma; BL, Burkitt lymphoma cancer; and BRCA, breast cancer. Note: (i) two databases have Tanzanian patients diagnosed with ESCC, and (ii) the BLGSP study cohort consists of subjects no more than 15 years old except for one at age 19 and is therefore defined as paediatric Burkitt lymphoma.

**Figure 2 cancers-17-02481-f002:**
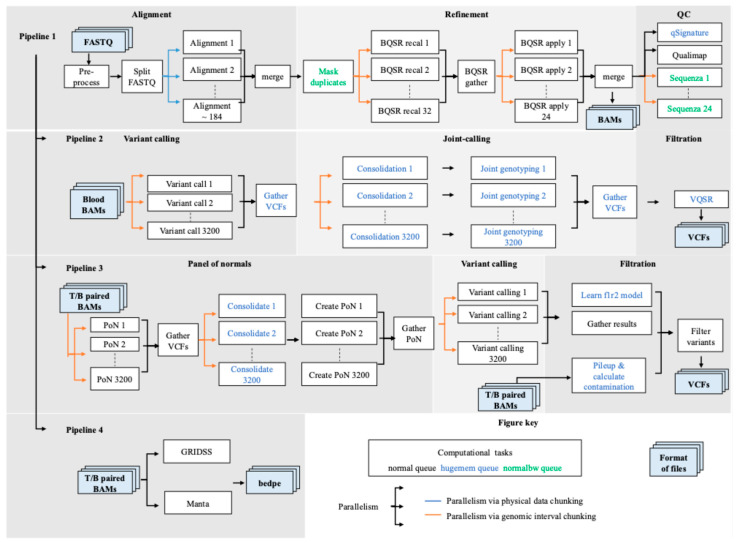
Schematic SAPCS workflow for processing African-inclusive cancer genomics data from WGS data to variant calling. The workflow is broken down into four pipelines including Pipeline 1, data processing and alignment; Pipeline 2, germline short variant calling; Pipeline 3, somatic short variant calling; and Pipeline 4, somatic structural variant calling. Files shown in blue boxes are inputs and outputs of computational tasks denoted as white boxes. Each task processed on National Computational Infrastructure (NCI) facilities is assigned with the optimised queue type, either normal queue in black, hugemem queue in blue, or normalbw queue in green. The sequential order of processing tasks is indicated by arrows. High-level parallel tasks are denoted by multiple arrows, with parallelism strategies indicated by colours, either via physical data chunking in blue colour or via genomic interval chunking in orange.

**Figure 3 cancers-17-02481-f003:**
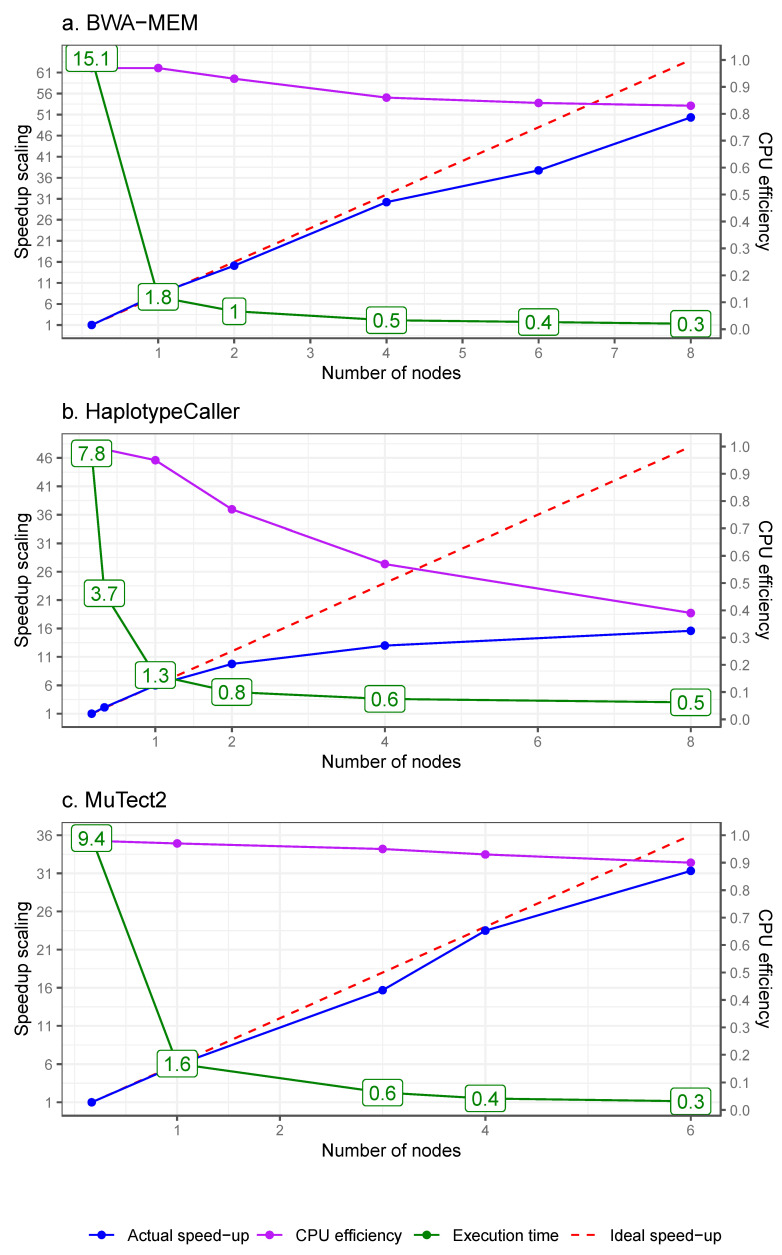
Scaling tests on computationally intensive analyses using African data from SAPCS. (**a**) The scalability of BWA-MEM was tested by aligning reads from a blood sample (30X coverage, 184 parallel tasks) with allocation of one to eight nodes (48 CPUs per node). Each parallel task was allocated six CPUs. (**b**) HaplotypeCaller was tested to call germline variants of a blood sample (3200 parallel tasks) with allocations from one to eight nodes. Each task is allocated one CPU. (**c**) Mutect2 was tested to process a blood sample (3200 parallel tasks) with allocations from one to six nodes. While the ideal speed-up scales linearly with the number of CPUs, the actual speed-up is defined as the product of execution time and CPU count for each process, compared to that of the process with the lowest CPU allocation. The lowest CPU allocation is six for BWA-MEM and eight for HaplotypeCaller and Mutect2. Performance is estimated as the inverse of the actual speed-up. CPU efficiency is an estimate of CPU time divided by the execution time and CPU count.

**Figure 4 cancers-17-02481-f004:**
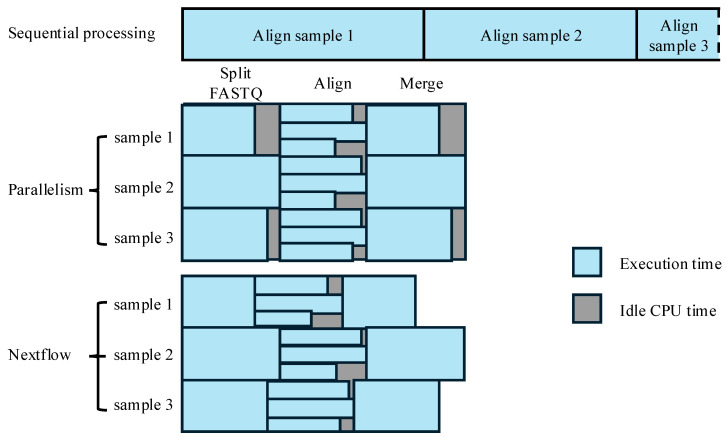
Schematic comparisons of African data processing in the alignment step using sequential processing, simple parallelisation, and automated workflow. The CPU time, in execution or idle, is denoted as boxes in blue and grey, respectively. For the sequential processing, a truncated box is indicated by a dashed line on the right-hand side as the full duration is not shown.

**Figure 5 cancers-17-02481-f005:**
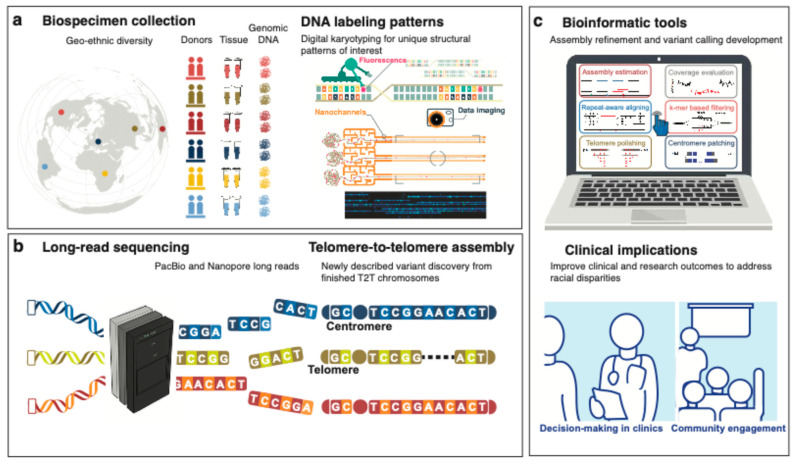
Overarching schematic of cancer genomics with a focus on diversity and inclusion. (**a**) Starting with the global collection of biospecimens chosen with unique patterns of digital karyotyping, as exemplified by Chan et al. [68]. (**b**) An innovative reference-free cancer genomics pipeline can be built based on the foundation of long-read technologies and telomere-to-telomere assembled chromosomes, followed by (**c**) a new ecosystem of analysis tools for their use in clinical applications.

**Table 1 cancers-17-02481-t001:** Cohort information of African WGS datasets of tumour and matched-normal tissue.

Consortium or Project	Cancer Type	Country	Cohort Size ^a^	Tissue Fixation ^b^	Coverage of Tumour, Normal (Median/Mean)	Recruitment Time	Recruitment Hospitals
SAPCS [26,27]	PCa	South Africa	123	FF	88.69X, 44.3X (median)	2013–2018	Polokwane Urology Clinic, Limpopo; Tshilidzini Hospital, Limpopo; Pretoria’s Steve Biko Academic Hospitals, Gauteng; Dr George Mukhari Academic Hospitals, Gauteng; and Kalafong Academic Hospital, Gauteng
ESCCAPE [28]	ESCC	Kenya	68	FF	49X, 26X (mean ^c^)	2014–2020	Moi Teaching and Referral Hospital, Eldoret;
		Malawi	59				Queen Elizabeth Central Hospital, Blantyre;
		Tanzania	35				Kilimanjaro Clinical Research Institute, Moshi
AfrECC [29]	ESCC	Tanzania	61	FFPE	60X, 30X (targeted coverage, de facto values unavailable)	2016–2018	Muhimbili National Hospital, Dar es Salaam,
BLGSP [30,31]	BL	Uganda	87	83 FF, 4 FFPE	82X, 41X (mean ^c^); 72.6X (mean across sample types ^c^)	Unavailable	Uganda Cancer Institute, Kampala; St Mary’s Hospital, Gulu
NBCS [32]	BRCA	Nigeria	97	FPAX	103.2X, 35.1X (mean)	2013–2015	Lagos State University Teaching Hospital, Lagos

Note: ^a^ cohort size: the number of cancer patients whose tumour and matched blood/normal samples underwent WGS; ^b^ FF: fresh frozen tissue; FPAX: Fresh PAXgene; FFPE: formalin-fixed paraffin-embedded tissue; ^c^ the mean coverage is calculated from the whole study cohort that include patients outside Africa.

**Table 2 cancers-17-02481-t002:** Main ancestral-related findings from African WGS datasets.

Cancer Type	Measurement	Values or Odds Ratios	*p*-Value	Comparison ^b^
Short variants (nucleotide variants, insertion and deletion variants less than 50 bp)				
PCa	Tumour mutational burden (TMB, mutations per Mb)	1.197 versus 1.061	0.013	EUR
PCa	Predicted damaging mutations (count)	14 versus 11	0.022	EUR
BRCA	Insertions and deletions (indels)	N/A	6.5 × 10^−5^, 2 × 10^−4^	EUR, AA
Driver genes				
BRCA	*GATA3*	6.3-fold	FDR = 0.038	EUR, AA
BRCA	Non-coding region, upstream of *ZNF217* (frequency)	42.3% versus 4.3%	FDR = 0.037	EUR, AA
BRCA	Non-coding region, spanning *SYPL1* (frequency	28.9% versus 0%	FDR = 0.097	EUR, AA
ESCC	*TP53* (frequency)	72% versus 74.8–87% [34,35,36]	-	EUR, AA
BL	*SIN3A* (frequency)	18.4% versus 9.1%	-	patients from the USA
BL	*HIST1H1E* (frequency)	9.2% versus 4.5%	-	
BL	*CHD8* (frequency)	9.2% versus 4.5%	-	
Somatic copy number alteration (SCNA)				
PCa	Percentage of genome alteration (PGA)	7.26% versus 2.82%	0.021	EUR
BRCA	Whole-genome duplications (WGD)	3-fold	FDR = 0.02	EUR, AA
Structural variants (SV)				
PCa	Duplication (relative frequency, count) [37]	1.6-fold, 2.5-fold	-	EUR
PCa	A single type hyper-SV frequency [37] ^a^	2-fold	-	EUR
PCa	*PCAT1*	9.09-fold	0.012	EUR
PCa	*TMPRSS2* *–* *ERG*	0.26-fold	0.0004	EUR
Several types of variants combined				
BRCA	intra-tumoral heterogeneity (ITH, increase %)	3.4%, 5.7%	0.005, 0.00017	EUR, AA
PCa	*NCOA2*	5.81-fold	3.14 × 10^−6^	EUR
PCa	*DDX11L1*	4.17-fold	0.0001	EUR
PCa	*STK19*	4.65-fold	0.004	EUR
PCa	*SETBP1*	2.80-fold	0.012	EUR

Note: ^a^ A single-type hyper-SV is defined as a tumour with at least 100 SVs dominated by a single type; ^b^ EUR, AA means significant comparisons between African patients with European, and African American patients, respectively.

**Table 3 cancers-17-02481-t003:** Bioinformatic tools applied to African WGS short-read data.

Consortium or Project	Genome	Variant Callers		
		Short Variants		Structural Variants
		Germline	Somatic	
SAPCS	GRCh38	GATK HaplotypeCaller [42]	GATK MuTect2 [43]	GRIDSS [44], Manta [45]
ESCCAPE	GRCh37	Strelka2 [46]	Strelka2, and cgpCaVEMan [47] for SNVs; cgpPindel [48] for INDELs	BRASS ^a^
AfrECC	GRCh37	-	RADIA [49]	-
BLGSP	GRCh38	-	Strelka2, GATK Mutect2, Lofreq [50], and SAGE ^b^	GRIDSS, Manta
NBCS	GRCh37	Platypus [51]	GATK MuTect and Strelka [52]	Manta, DELLY [53], and Lumpy [54]

Note: ^a^ https://github.com/cancerit/BRASS (accessed on 9 June 2025), ^b^ https://github.com/hartwigmedical/hmftools/blob/master/sage (accessed on 9 June 2025).

**Table 4 cancers-17-02481-t004:** Configurations for SAPCS workflow compute jobs. Estimates of data processing with a batch of 20 pairs of tumour and matched-blood samples using National Computational Infrastructure (NCI) facilities.

Steps	Sample Type ^a^	CPU/Task	Total Tasks	Batches	CPUs/Batch	Execution Time (h)	Main Algorithm with Version
Pipeline 1 Data pre-processing for variant discovery						14.4	
Split FASTQ	Bood	4	20	1	96	0.9	fastp [58] v0.20.0
	Tumour	4	20	1	96	1.8	
Alignment	Both	6	11,040	3	3840	0.5	BWA-MEM v0.7.15
Merge	Bood	24	20	1	480	0.4	SAMBAMBA [59] v0.7.1
	Tumour	24	20	1	480	0.8	
Mask duplicate	Bood	14	20	1	280	1.3	SAMBLASTER [60] v0.1.24
	Tumour	14	20	1	280	2.6	
BQSR recal	Bood	1	640	1	640	0.2	GATK v4.4.0.0 ^b^ BaseRecalibrator
	Tumour	1	640	1	640	0.3	
BQSR apply	Bood	2	480	1	960	0.3	GATK ApplyBQSR
	Tumour	2	480	1	960	0.6	
qSignature	Bood	24	20	1	480	0.7	QSignature ^c^ v0.1pre (75)
	Tumour	24	20	1	480	1.4	
Qualimap	Bood	6	20	2	144	1.4	Qualimap [61] v.2.2.1
	Tumour	6	20	2	144	2.8	
Sequenza	Pair	2	480	1	504	3.6	Sequenza [62] v3.0.0
Pipeline 2 Germline short variant discovery						8.1	
Variant call	Bood	1	64,000	1	480	1.8	GATK HaplotypeCaller
Consolidation	Bood	1	3200	11	144	1.3	GATK GenomicsDBImport
Joint genotyping	Bood	1	3200	1	144	2	GATK GenotypeGVCFs
VQSR	Blood	16	1	1	16	3	GATK VariantFiltration, MakeSitesOnlyVcf, VariantRecalibrator, CollectVariantCallingMetrics, ApplyVQSR, CollectVariantCallingMetrics
Pipeline 3 Somatic short variant discovery						3.3	
PoN	Bood	1	64,000	1	2880	0.6	GATK Mutect2
Consolidate	Blood	2	3200	1	96	0.3	GATK GenomicsDBImport
Create PoN	Blood	1	3200	1	960	1.6	GATK CreateSomaticPON
Variant call	Pair	1	64,000	1	2880	0.8	GATK Mutect2
Pipeline 4 Structural variant discovery						23	
GRIDSS	Pair	8	20	20	8	Range, 10–20	GRIDSS v2.8.3
Manta	Pair	24	20	2	48	3.0	Manta v1.6.0

Note: ^a^ Both means that tumour and blood samples are processed in one job but as separate tasks. Pair means that tumour and the matched blood are processed together in one task. Steps performing high-level parallelism are highlighted in grey. Small steps processing for a few minutes is omitted. ^b^ GATK tools are all v. 4.4.0.0, ^c^ https://github.com/AdamaJava/adamajava/tree/master/qsignature (accessed on 9 June 2025).

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
