# Peer review of "Scaling for African Inclusion in High-Throughput Whole Cancer Genome Bioinformatic Workflows"

_cancers, 2025, doi:10.3390/cancers17152481_

Round 1
Reviewer 1 Report
Comments and Suggestions for Authors
This study presented a systematic review of African-inclusive whole cancer genome studies (WGS), addressing a critical gap in cancer genomics research: the lack of African representation and inefficient of processing pipelines developed primarily for non-African populations. The authors discussed computational and data-related challenges associated with African WGS from several aspects. The authors provided a rationale for selecting bioinformatic workflows that support parallel processing of complex, high-intensity data and improve variant calling accuracy.
Overall, it is a valuable review and comparative study that aims to provide a standardized, practical bioinformatic guideline for African cancer research. It would serve as a useful reference for bioinformatics practitioners and researchers working in the field. However, there are several aspects that authors could add or elaborate on to enhance the impact and clarity of the manuscript.
- The author discussed computational challenges and proposed a rationale of bioinformatic pipeline. However, the study lacks quantitative performance benchmarks to support the claims of improved scalability and efficiency comparing to other pipelines. Including direct performance comparisons would provide stronger evidence for the advantages of the proposed workflow.
- Similarly, authors claimed that the proposed pipeline improves variant calling accuracy, but no supporting data were provided. To validate this claim, a numerical comparative analysis of accuracy metrics between the proposed pipeline and existing standard workflows should be included.
- This manuscript would be significantly strengthened by providing accessible software or making the pipeline and associated code publicly available. Doing so would greatly enhance the manuscript’s value, reproducibility, and impact within the research community.
Author Response
Reviewer 1
This study presented a systematic review of African-inclusive whole cancer genome studies (WGS), addressing a critical gap in cancer genomics research: the lack of African representation and inefficient of processing pipelines developed primarily for non-African populations. The authors discussed computational and data-related challenges associated with African WGS from several aspects. The authors provided a rationale for selecting bioinformatic workflows that support parallel processing of complex, high-intensity data and improve variant calling accuracy.
Overall, it is a valuable review and comparative study that aims to provide a standardized, practical bioinformatic guideline for African cancer research. It would serve as a useful reference for bioinformatics practitioners and researchers working in the field. However, there are several aspects that authors could add or elaborate on to enhance the impact and clarity of the manuscript.
Response: We appreciate the acknowledgement for the value of this timely review. Here, we will further address the specific areas raised to ensure we enhance the clarity and impact of the paper.
- The author discussed computational challenges and proposed a rationale of bioinformatic pipeline. However, the study lacks quantitative performance benchmarks to support the claims of improved scalability and efficiency comparing to other pipelines. Including direct performance comparisons would provide stronger evidence for the advantages of the proposed workflow.
Response: A paragraph was added to the beginning of Section 3.2 to show the improved execution time for the pipeline after applying high-level parallelism. As such, we compared the processing of somatic variant calling with the pipeline conducted by the Pan Prostate Cancer Group (PPCG) consortium tested on seven African samples. The additional paragraph reads as follows:
For African ancestry WGS presenting elevated germline and somatic variants, its workflow needs to be scalable for improved execution time efficiency and, therefore, the analysis of a larger cohort size. The pipelines described above have been tailored to accommodate two types of high-level parallelism: physical data chunking for read alignment and the scatter-gather of genomic interval processing for variant calling. The execution time of these steps has been improved substantially. Using hundreds of computational cores, the execution time of the alignment has been reduced from 15.1 hours to 2 hours for a 30X sample, and the germline haplotyping step improved from 7.8 hours to 0.5 hours. The somatic variant calling pipeline took approximately 3.3 hours for a cohort of 20 patients, compared to 36 to 47 hours using the pipeline employed by the Pan Prostate Cancer Group (PPCG) consortium (https://github.com/cancerit/dockstore-cgpwgs; 48 CPUs and 960 GB each).
We still acknowledge that the current proposed pipeline is not the fastest and could be further improved, as originally illustrated in Section 3.3.
- Similarly, authors claimed that the proposed pipeline improves variant calling accuracy, but no supporting data were provided. To validate this claim, a numerical comparative analysis of accuracy metrics between the proposed pipeline and existing standard workflows should be included.
Response: For the variants calling pipelines, we acknowledge the limitation of not having data to support improved accuracy for the GATK pipeline. As this is a review paper rather than a research article, we looked for previous publications on this topic. However, current evaluations on accuracy between tools/pipelines used European individuals, most commonly NA12878. We found no such “truth set” for African subjects. Also, the evaluation of accuracy is often on a sample level, not on a cohort level, making the joint-calling function of the GATK pipeline futile. As such, we removed the statement emphasising accuracy and added this limitation to Section 5.
To avoid leading to the comparison of pipelines, the following minor changes (in red) were made to the text and include:
“Reviewing bioinformatic tools applied to African databases, we carefully select a workflow suitable for a large-scale African database.”
“Such demands increase proportionally with cohort size, incurring additional computational costs associated with achieving greater statistical power and sensitivity.”
“The following refinement, which facilitates the utmost sensitivity and specificity of variant calling, includes masking duplicate reads (technical artefacts that may cause false positives) utilising SAMBLASTER and GATK base quality score recalibration (BQSR) to lessen the impact of systematic errors introduced during sequencing.”
“The strategy is applied in Pipelines 1-3 with varying numbers of genomic intervals to optimise execution time and computational load without impacting outcome.”
The limitation added in Section 5:
“We acknowledge that future studies are required to determine variant calling accuracy of the tested pipeline. The analysis should compare the performance on a "truth set" of African ancestry and test the improvements of integrating cohort information. “
- This manuscript would be significantly strengthened by providing accessible software or making the pipeline and associated code publicly available. Doing so would greatly enhance the manuscript’s value, reproducibility, and impact within the research community.
Response: We agree this is very important. In Section 3.1, we added a clearer note to specify that codes of scripts are available online by checking the references from No. 56 to 58.
“SAPCS applied a parallelism-integrated workflow (code/scripts available online)56-58”
“(56) Sadsad, R., Samaha, G., & Chew, T. . Fastq-to-bam @ NCI-Gadi. WorkflowHub 2021. DOI: https://doi.org/10.48546/WORKFLOWHUB.WORKFLOW.146.1.
(57) Sadsad, R., Samaha, G., & Chew, T. . Germline-ShortV @ NCI-Gadi. WorkflowHub 2021. DOI: https://doi.org/10.48546/WORKFLOWHUB.WORKFLOW.143.1.
(58) Sadsad, R., & Chew, T. Somatic-ShortV @ NCI-Gadi. WorkflowHub 2021. DOI: https://doi.org/10.48546/WORKFLOWHUB.WORKFLOW.148.1.”
Reviewer 2 Report
Comments and Suggestions for Authors
In this study, the authors highlight that Sub Saharan Africa bears the highest global cancer mortality, yet its populations remain largely excluded from precision oncology initiatives. A review of five comprehensive whole-cancer genome databases—including breast, esophageal, prostate cancers, and Burkitt lymphoma—revealed markedly elevated genomic instability in African-derived tumours, characterized by heightened tumour mutational burden, structural variants, copy-number alterations, and frequent whole-genome duplications. These studies also identified novel, African-specific oncogenic drivers (e.g., NCOA2, STK19, SETBP1, PCAT1, DDX11L1) and distinct mutational signatures, pointing to unique biological mechanisms at play. Critically, the complexity of these genomes underscores limitations in conventional bioinformatic pipelines, prompting the authors to advocate for scalable, parallelized workflows—such as genomic interval chunking—augmented by African-derived reference data and multi‑platform sequencing to enhance variant-calling accuracy. These targeted computational strategies aim to advance early diagnosis and personalized treatment, ultimately helping to close the cancer mortality disparity in Sub‑Saharan Africa.
This strong, well written and perfectly done study can be published in Cancers. There are only several minor points:
Abstract: Please, clarify the objectives, hypotheses, and research questions early in the text. Describe the tools/workflows in more detail. Please, underline the novelty of the study.
Discussion:
Could you elaborate on the underlying biological, genetic, and environmental factors that may contribute to the increased tumor genome instability and the emergence of African-specific cancer driver mutations observed in Sub-Saharan African populations? It would be good to discuss Biological mechanisms (e.g., DNA repair defects, mutational processes), Population genetics (e.g., higher genetic diversity, underrepresented reference genomes), Environmental and lifestyle exposures (e.g., infectious agents, dietary factors, carcinogen exposure), Healthcare disparities (e.g., late diagnosis, lack of screening), which all may play a role in the observed genomic patterns.
It also would be good to outline perspectives and limitations of the study.
Reviewer 3 Report
Comments and Suggestions for Authors
While the manuscript underscores the importance of scalable bioinformatic workflows for analysing African cancer genome data, several key computational details remain unclear and warrant clarification to enhance reproducibility and technical applicability:
1. Please define "scalability" more precisely. Is it measured by CPU time, memory usage, throughput, or cross-platform compatibility? Clarifying this will help readers interpret the pipeline benchmarking more accurately.
2. Please elaborate on the parallelisation strategy used in the SAPCS benchmark. Were workflows distributed across multiple cores at the sample level, per pipeline step, or via scatter-gather? This information is essential for replication in diverse computing environments.
3. Please discuss the suitability of SV callers for African genomes. Given the high structural variant burden, are current tools (e.g., Manta, Delly, GRIDSS) sufficient, or would graph-based or long-read–supported methods offer advantages?
4. Please comment on population-specific PoN construction. Are African-specific PoNs necessary to avoid false positives? If so, what is the recommended minimal number of normals for stable modelling?
5. Please advise on best practices for tumour-only variant calling in high-diversity genomes. Would the authors recommend using African-specific allele frequency filters or machine learning classifiers to improve somatic call accuracy in the absence of matched normals?
6. Please expand on the feasibility of using African-centric or graph-based genome references. Are current pipelines compatible with pangenomic references such as AGVP or T2T builds, and what are the main technical challenges to integration?
